# Gender-Specific Effects of 8-Week Multi-Modal Strength and Flexibility Training on Hamstring Flexibility and Strength

**DOI:** 10.3390/ijerph192215256

**Published:** 2022-11-18

**Authors:** Shangxiao Li, Liduan Wang, Jinfeng Xiong, Dandan Xiao

**Affiliations:** 1Research Center for Sports Psychology and Biomechanics, China Institute of Sport Science, Beijing 100061, China; 2School of Rehabilitation Medicine, Weifang Medical University, Weifang 261053, China; 3Institute of Sports Science, Sichuan University, Chengdu 610065, China

**Keywords:** strength, flexibility, multi-modal training, injury prevention

## Abstract

The purpose of this study was to investigate the effects of multi-modal strength training or flexibility training on hamstring flexibility and strength in young males and females. A total of 20 male and 20 female college students (aged 18–24 years) participated in this study and were randomly assigned to either a multi-modal flexibility intervention group or strength intervention group. Passive straight leg raise and isokinetic strength test were performed before and after the intervention to determine flexibility and strength of the participants. Multivariate repeated-measure ANOVA was used to determine the effect of training group and gender on hamstring strength and flexibility. Both male and female participants in the strength intervention group significantly increased peak torque, relative peak torque, and flexibility (all *p* ≤ 0.029). Both male and female participants in the flexibility intervention group significantly increased flexibility (both *p* ≤ 0.001). Female participants in the flexibility intervention group significantly increased peak torque and relative peak torque (both *p* ≤ 0.023). However, no change was seen in peak torque and relative peak torque of male participants in the flexibility intervention group (*p* ≥ 0.676). An 8-week strength training program involving various training components can increase flexibility in both males and females, although the flexibility of male participants only increased slightly. While hamstring flexibility training protocol consisted of different types of stretching improved both flexibility and strength in female participants, male participants increased only flexibility but not strength, indicating such effects were gender-specific. For subjects with relatively low strength (e.g., older adults, sedentary women, postoperative rehabilitation population, etc.), strength training alone or flexibility training alone may increase both strength and flexibility.

## 1. Introduction

Strength and flexibility are among the most important physical fitness features [1]. Strength and flexibility help people attain a healthy physical fitness level and functional autonomy, and play key roles in reducing sports injuries and improving sports skills and ability [1,2].

Strength training alone may lead to an increase in flexibility. A study completed by Santos et al. showed that an 8-week moderately intense strength training can improve shoulder and trunk joint flexibility in young sedentary women [3]. Barbosa et al. demonstrated that a 10-week weight training without performing stretching exercises significantly increased sit-and-reach test score of elderly women [4]. Further, several studies have shown that strength training alone over a period of time promotes flexibility in older adults [5,6]. In addition, several studies concluded that eccentric training programs increased joint flexibility [7,8], supporting that eccentric exercise is an effective method of increasing lower limb flexibility [9].

While most studies have shown acute decreases in strength after stretching [10,11], the literature on the chronic effects of flexibility training on muscle strength showed strength may increase [12,13] or not change [14,15]. A study by Behm et al. recruited twelve male college students to participate in a stretching training program on quadriceps and hamstring muscle, and the results showed a significant decrease (−6.1% to −10.7%) in muscle strength after an acute bout of static stretching while no significant decrement on strength was observed after 4 weeks of flexibility training [14]. In addition, it was shown that 8 weeks of proprioceptive neuromuscular facilitation (PNF) stretching increased muscle strength of knee flexors (15.5%) and knee extensors (6.1%) [13], and static stretching increase muscle strength of knee flexors in untrained young men (8.7%) [13]. Several existing studies supported that flexibility training alone over a period of time promotes strength [12,16].

Although the strength or flexibility training protocols used in actual practice usually incorporate multiple training modalities, researches on the relevant topic generally focused on the effects of a single type of technique [8,12] whose results may not be conveyed to daily training. Multi-modal training may show superiority over one single training method. Further examination of the influence of multi-modal strength and flexibility training on flexibility or strength is an important part towards a comprehensive understanding of the synergistic (or counteractive) effects of different training modalities, which could be important when prescribing physical exercise programs with the purpose to increase the effectiveness of training. Such information is also crucial for training programs aimed at either preventing sports-related injuries or optimizing recovery after injuries occur.

Gender may play an important role in the physical fitness improvement in response to different types of training. Beneka et al. found that resistance training effects were related to gender [17]. Thomas et al. also found that the effects of high-velocity strength training on isometric strength were gender specific [18]. While training produced improvements in counter movement jump height in both genders, significant increases in maximal isometric force and rate of force development were found only in females [18]. However, most studies focused on single gender participants and did not consider gender as a potential confounding factor [6,12,14], while some studies analyzed the results of both genders as one group [16].

Therefore, the purpose of this study is to investigate how strength training or flexibility training protocol that incorporates multiple training modalities can influence flexibility and strength for males and females. Hamstring muscle was studied as hamstring strain injury is one of the most common injuries with serious consequences in sports [19], and flexibility and strength were the most investigated modifiable risk factors for hamstring injury [20]. We hypothesized that (1) 8 weeks of multi-modal strength training can significantly improve hamstring strength in young healthy male and female adults without reducing hamstring flexibility, and (2) 8 weeks of multi-modal flexibility training can significantly improve hamstring flexibility in young healthy male and female adults without reducing strength.

## 2. Materials and Methods

### 2.1. Participants

Forty healthy college students (aged 18–24 years, 20 males and 20 females) regularly participating in running or basketball, 2–3 times a week, volunteered for this study and were randomly assigned to either a strength intervention group or a flexibility intervention group. Using a block randomization procedure with a block size of 4, participants were randomly assigned to either a flexibility intervention group or a strength intervention group. A block was randomly picked by the 2th author (LDW) for every 4 incoming participants. The assignment of each participant was revealed after the participant signed the consent form. All participants had no history of lower extremity injuries 2 years prior to the study. This study was approved by the Institutional Review Board. Prior to any data collection, all participants signed a consent form. Detailed information for all participants was shown in Table 1.

### 2.2. Overview of Procedures

All participants completed a baseline testing, an 8-week multi-modal strength training intervention or flexibility training intervention, and a post-intervention testing session. All participants were trained and tested bilaterally. The pre- and post-intervention testing sessions started with a passive straight leg raise (PSLR) test for flexibility evaluation, and an isokinetic strength test for maximum muscle strength evaluation. Participants were instructed not to perform any vigorous physical activity at least 48 h prior to either testing session. All tests were carried out in the same laboratory, with room temperature and humidity set at the same.

### 2.3. Training Session Protocol

All participants were trained three times per week for 30 min per session. A minimum of 36 h between two consecutive sessions were given to provide enough recovery time for the participants. Each training program consisted of 6 min of standardized warm-up including jogging and reverse lunge and heel to butt exercises, followed by a series of hamstring flexibility or strength training intervention for about 25 min, and was supervised by certified personal trainers. The strength intervention included a series of knee flexion exercises with resistance and Nordic hamstring curl exercises, with details shown in Table 2. The flexibility intervention included a series of dynamic, static, and PNF stretches, with details shown in Table 2. The type of PNF employed in the current study was based on the contract-relax technique [21], which begins with a passive pre-stretch of the hamstrings that is held at the point of mild discomfort for 10 s, then the participant extends the hip against resistance from the trainer for 6 s. The participant then relaxes, and a passive hip flexion stretch is applied and held for 10 s [21]. One set of PNF stretch includes repeating the abovementioned procedure twice. Before the training program training starts, all trainers and participants received written descriptions and illustrations of the required interventions. All trainers were trained and examined to ensure that required exercises were performed appropriately. The participants were instructed to maintain their current physical activity level and dietary routine, and avoid any structured exercises while participating in the study.

### 2.4. PSLR Test for Flexibility

The flexibility of all participants was assessed three times using the PSLR test combined with a high-definition digital camera (GC-PX100, JVC, Yokohama, Japan) placed perpendicularly to the participant’s sagittal plane to record body position. All tests were administered by the same experienced tester to avoid intra-operator variability. The test started with the participant laid supine on a cushion, and the tester placed one hand on the superior iliac crest of the non-testing side of the participant to prevent the pelvis from rotating, while lifting the testing leg slowly using the other hand. The participant’s knee was kept extended throughout the test. The PSLR test was terminated when the participant’s pelvis started to rotate, or when he/she started to feel discomfort or large resistance that caused the testing leg no longer able to be lifted. Each participant was tested three times for each leg, with 60 s rest time given between two consecutive tests.

### 2.5. Isokinetic Strength Test for Strength

Hamstring strength was assessed by measuring knee flexion torque using an IsoMed 2000 strength testing dynamometer (D&R Ferstl GmbH, Hemau, Germany). The participant sat on the dynamometer with hip fixed at 90°, and the torso and pelvis secured on the testing chair. The knee joint of the testing leg was aligned with the rotator arm of the dynamometer, and the range of motion (ROM) for the rotator arm was between 0–110°, with 0° being when the knee was fully extended. The angular speed of isokinetic testing was set at 10°/s. The participant was instructed to perform the isokinetic test with their maximal effort, and each participant was tested three times for each leg, with 90 s rest time given between two consecutive tests.

Data of this study was previously presented to determine the effects of altering hamstring flexibility or strength on hamstring optimal lengths [22]. However, in this manuscript, we sought to discuss whether multi-modal hamstring flexibility and strength training were gender-specific.

### 2.6. Data Analysis

Maximal hip flexion angle was determined as the maximal hip flexion angle from the digitized photo taken during the PSLR test. The range of hip joint flexion in each PSLR trial was calculated as the angle between the vector from the hip joint center to knee joint center, and the vector from the acromion process to hip joint center. The average of the maximal hip flexion angles from three PSLR trials was used as the hamstring flexibility score for each leg.

Isokinetic strength data were processed on the dynamometer to get peak torque of the hamstring during the isokinetic test. Relative peak torque was calculated as the peak torque divided by body mass, with a unit of Nm/kg. The strength testing trial that had the maximal peak torque and relative peak torque was selected to represent hamstring strength.

### 2.7. Statistical Analyses

Multivariate repeated-measure ANOVA was used to determine the effect of training group and gender on hamstring strength and flexibility, with time as a factor in the repeated-measure. If the three-way interaction was significant, then additional analysis was carried out as needed and is detailed in the Section 3. If the three-way interaction was not significant, then simple effects (LSD) test was used.

All statistical analyses were performed using Version 18.0 of SPSS computer program package (SPSS, Chicago, IL, USA). Statistical significance was defined as the type I error rate lower than or equal to 0.05. To measure the magnitude of training effects, Cohen’s d (d) was calculated when applicable, with 0.2, 0.5 and 0.8 considered as small, medium and larger effects, respectively.

## 3. Results

All participants completed all training sessions. The three-way interaction from the multivariate repeated-measure ANOVA was not significant for peak torque or relative peak torque, so simple effect was used for subsequent tests. The three-way interaction for flexibility was significant, and further analyses revealed that the two-way interaction between time and gender, as well as between time and group, was significant. Paired t-test (two-tail) was used for post-hoc analysis accordingly.

Peak torque of male participants and female participants in the strength intervention group were both significantly increased in the post-intervention test in comparison to the pre-intervention test (*p* = 0.010 and 0.004, respectively; Figure 1) with large effect size (*d* = 0.80 and 0.99, increased by 9.0% and 16.3%, respectively). Relative peak torque of both male and female participants also increased (*p* = 0.009 and 0.001, respectively; Figure 1) with large effect size (*d* = 0.80 and 1.00, increased by 9.0% and 16.6%, respectively). Flexibility of male participants as well as female participants in the strength intervention group were significantly increased in the post-intervention test in comparison to the pre-intervention test (*p* = 0.029 and <0.001, respectively; Figure 1) with small and large effect size (*d* = 0.43 and 1.52, increased by 4.5% and 10.5%, respectively).

Peak torque of female participants in the flexibility intervention group was significantly increased in the post-intervention test in comparison to the pre-intervention test (*p* = 0.023; Figure 2) with medium effect size (*d* = 0.67, increased by 11.4%), while peak torque of male participants was not significantly changed in the post-intervention test (*p* = 0.676, *d* = −0.06; Figure 2). Relative peak torque of female participants in the flexibility intervention group was significantly increased in the post-intervention test in comparison to the pre-intervention test (*p* = 0.009; Figure 2) with medium effect size (*d* = 0.66, increased by 11.0%), while peak torque of male participants was not significantly changed in the post-intervention test (*p* = 0.803, *d* = −0.04; Figure 2). Flexibility of male participants as well as female participants in the strength intervention group were significantly increased in the post-intervention test in comparison to the pre-intervention test (both *p* < 0.001; Figure 2) with large effect size (*d* = 2.06 and 3.10, increased by 18.1% and 19.0%, respectively).

## 4. Discussion

Results from the current study suggest that a multi-modal strength training intervention program consisting of concentric, eccentric, and isometric training can significantly increase hamstring peak torque and relative peak torque in both males and females (*d* ≥ 0.80 for both groups), which is consistent with our hypothesis. Therefore, we demonstrated that an 8-week integrated strength training protocol employed in this study can be used to increase hamstring strength. In addition, significant increase in the PSLR test score was also observed for both male and female participants (*p* ≤ 0.029, *d* ≥ 0.43), although flexibility of males only increased by 4.5%. This is consistent with our first hypothesis that 8 weeks of multi-modal strength training does not reduce hamstring flexibility.

The results of our study support the notion that multi-modal strength training without stretching increases flexibility on healthy young adults. Several studies have found that strength training alone contributes to the development and maintenance of flexibility even without additional stretching [3,4,6,8,14]. However, these studies generally focused on the effects of strength training on aging population or sedentary participants [3,4,6]. Moreover, studies have shown that the flexibility improvement with strength training may be intensity-dependent and volume-dependent [6,23], with intensities 60% or 80% of 1RM are more effective in producing flexibility gains compared to intensities 40% of 1RM [6]. This implies that for active individuals who possess relatively high level of strength, higher training loads of strength training may be needed to increase flexibility. In addition, the strength training method of our study included a series eccentric training. It has been consistently shown that eccentric strength training could increase angular ROM and fascicle length for several muscles, including the quadriceps, hamstring and triceps surae [9]. It is worth mentioning that the effect size of flexibility gains by strength training was larger for females (*d* = 1.52) while being small for males (*d* = 0.43). The difference may come from training loads and gender differences. Male participants had greater strength before training intervention. While our training protocol mainly focused on self-weight training, the level of loading for males may be relatively lower compared to females.

The mechanism underlying the improved flexibility in response to strength training is not fully understood, but several possibilities have been proposed. The first is related to the physical property of the muscle. It has been shown that strength training could induce a decrease in stiffness of the active part of the series elastic component [24], reduce passive torque [7], and increase tensile strength of tendons and ligaments [25]. Second, eccentric exercise is more likely to generate passive stretch to elongate the muscle fibers [26]. An animal study showed that eccentric training induced longitudinal adaptation of the sarcomeres [27], which may increase muscle fascicle length and subsequently lead to improved flexibility. In the current study, we employed a strength protocol which encompasses concentric, eccentric as well as isometric training components. The enhanced flexibility seen in both male and female participants following the 8 weeks training is likely due to both the physical property changes in the hamstring in response to strength training, as well as the effects of stretch of the muscle fibers that is caused by eccentric exercise.

The results from the current study also suggest that a multi-modal flexibility training protocol that is consisted of static, dynamic, and PNF stretching can significantly improve hip ROM in both male and female participants (*p* < 0.001, *d* ≥ 2.00 for both groups), which is consistent with our second hypothesis and suggest that the flexibility training protocol employed in this study can be used to increase flexibility. No change for strength was observed in males (*p* ≥ 0.627, |*d*| ≤ 0.06), while significant increase in strength was seen for female participants with medium effect size (*p* ≤ 0.039, *d* ≥ 0.62). Therefore, the effects of flexibility training on strength seem to be gender specific.

Although there were some heterogeneity, most longitudinal studies found either increase or no change in muscle strength following flexibility training, mainly for the lower limb muscles [14,28,29]. Recently, a review of 25 studies indicated that flexibility training may increase strength as measured during isotonic contractions, but no improvement on strength was seen as measured using isometric contractions test [30]. That implied that different methods used to evaluate strength may also contribute to the inconsistent findings. Interestingly, study by Nelson et al. found that a 10-week unilateral static stretching intervention in college students with no training experience not only increased strength in triceps surae on the side being trained, but also showed a crossed training effect on the contralateral side, as its strength was also improved [16]. Another 8-week PNF training study also found a cross-training effects, which increased by 9.9% on knee extension torque on the non-training side [31]. This could be important information for rehabilitation programs aiming at improving muscle function of an immobilized leg. The results from the current study showed no strength change for males but significant increase for female participants by an 8-week multi-modal flexibility training. Our study indicates that long-term flexibility training may be an effective way to not only improve flexibility, but also simultaneously improve strength, although in a gender-specific manner. Clearly more research is needed to confirm this notion.

There are several possible mechanisms that may lead to the strength gain after longitudinal flexibility training. The first is the structural adaptations in skeletal muscle. Stretching has been shown to increase muscle mass, muscle length, muscle fiber size, serial sarcomere number, and cross-sectional area in quails or rats [32,33]. Further analysis also suggested increased muscle protein synthesis. These changes laid the structural foundation for the potential muscle strength increase [34]. The second possible mechanism is the alteration of physical properties of the muscle. It is possible that flexibility training induced skeletal muscle physical property changes. It was reported that flexibility training can increase the compliance of the series elastic component, reduce the viscoelastic properties of the tendon, and store more elastic energy [35]. These changes could build the physical foundation for strength gain in muscle. The third potential mechanism is the enhanced neuromuscular adaptation. A study by Yamashita et al. indicated that stretching can increase the Ca^2+^ conductance at nerve terminal, facilitate the release of neurotransmitters, and therefore improve neuromuscular transmission [36]. Coupled with other studies which indicated that flexibility training leads to enhanced neuromodulation and reflex sensitivity [16,37], it seems that a better neuromechanical adaptation is induced by this training modality that prepares for the possible strength gain in skeletal muscle. Last but not least, PNF stretching does include a strength training component. Several studies have reported that those with low level of strength at baseline could increase strength following PNF stretching, as this type of stretching includes both concentric and isometric contractions against external loading [29,31].

The results of our study suggest that the effect of the multi-modal flexibility training protocol employed in the current study on hamstring strength appears to be gender specific. One previous study also investigated the gender-related effects in strength training and concluded that females have higher adaptability following strength training, which is likely attributed to their lower absolute as well as relative strength than males [18]. Male participants in our study may have better neural recruitment pattern and the effect of training on nervous system was relatively low, therefore no strength gain was observed in male participants. As previously mentioned, strength gain by flexibility training was mostly studied in old or sedentary adults, and those are individuals considered to have low strength to begin with [16,28]. This study has shown that the increase in strength is mainly due to muscle hypertrophy and increased recruitment of neurons, and improvement in neuron recruitment generally happens at the early stage of training [38]. The hamstring strength gain seen in female participants in this study is primary due to the results of the enhanced neuromuscular conductance, as our training only lasted eight weeks. Taken together, the influence of flexibility training on strength appears to be gender specific, which is likely due to the different baseline strength between males and females. Therefore, those with low baseline strength may have better strength gain following flexibility training, although more specific research on this topic is required.

Studies on how strength training and flexibility training may influence strength and flexibility have yielded conflicting results. Two studies from the same research team included both flexibility training and strength training produced conflicting results [15,39]. One study drew the conclusion that flexibility training did not change strength while strength training increased flexibility [15]. However, findings from the other study showed that flexibility training increased strength slightly while strength training did not change flexibility [39]. A similar study concluded that increases in muscle strength and flexibility are only developed by specific training programs [40]. Yet, gender differences were not considered, although the study enrolled 35% female participants [40]. Different strength levels between males and females before training may partly explain the results in our study which showed that the effect of flexibility training on strength were gender-specific. To sum up, the inconsistent findings in the literature may be attributed to various factors, including but not limited to subject characteristics (such as gender, age and physical activity level), specific training contents (type of training, intensity and duration of training), as well as variety methods employed to evaluate flexibility and strength.

The limitations of this study must be addressed. First, muscle strength was tested as the peak torque during isokinetic strength test, which only represents one aspect of strength. In addition, the training protocols were designed for individuals who do not have professional training experience, which may not be suitable for athletes who already possess high level of strength and flexibility. Therefore, our findings may not be generalized into this population. Third, this study only evaluated strength and flexibility in response to training. How strength and flexibility training may influence other abilities crucial to movement, such as balance and agility may need to be further explored.

## 5. Conclusions

An 8-week strength training program involving various training components can increase flexibility in both males and females, although the flexibility of male participants only increased slightly. While hamstring flexibility training protocol consisted of different types of stretching improved both flexibility and strength in female participants, male participants increased only flexibility but not strength, indicating such effects were gender specific. When prescribing physical exercise programs, it is necessary to take gender or baseline muscle strength level into consideration. For subjects with relatively low strength (e.g., older adults, sedentary women, postoperative rehabilitation population etc.), strength training alone or flexibility training alone may increase both strength and flexibility.

## Figures and Tables

**Figure 1 ijerph-19-15256-f001:**
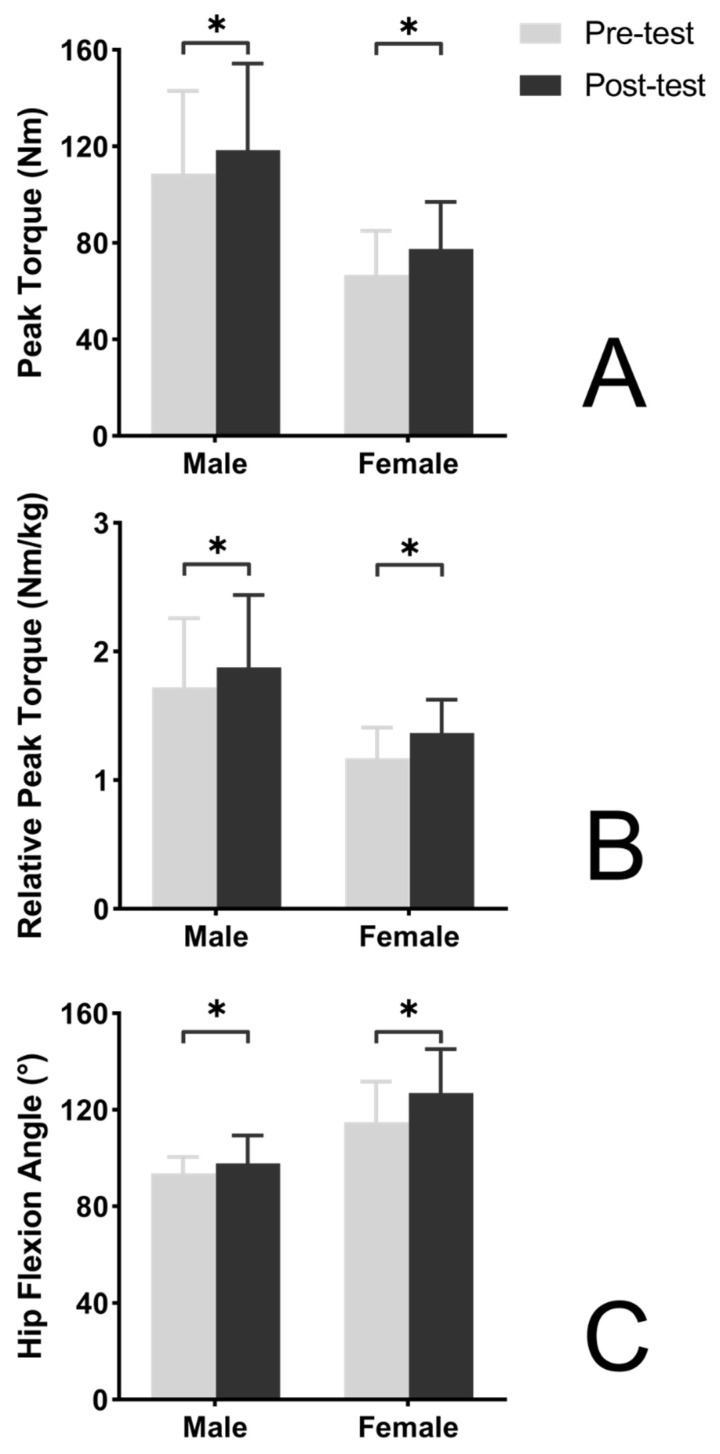
Effects of an 8-week strength training program on hamstring strength and flexibility. (**A**) Training effect on peak torque. (**B**) Training effect on relative peak torque. (**C**) Training effect on hip flexion angle. * Significant training effects.

**Figure 2 ijerph-19-15256-f002:**
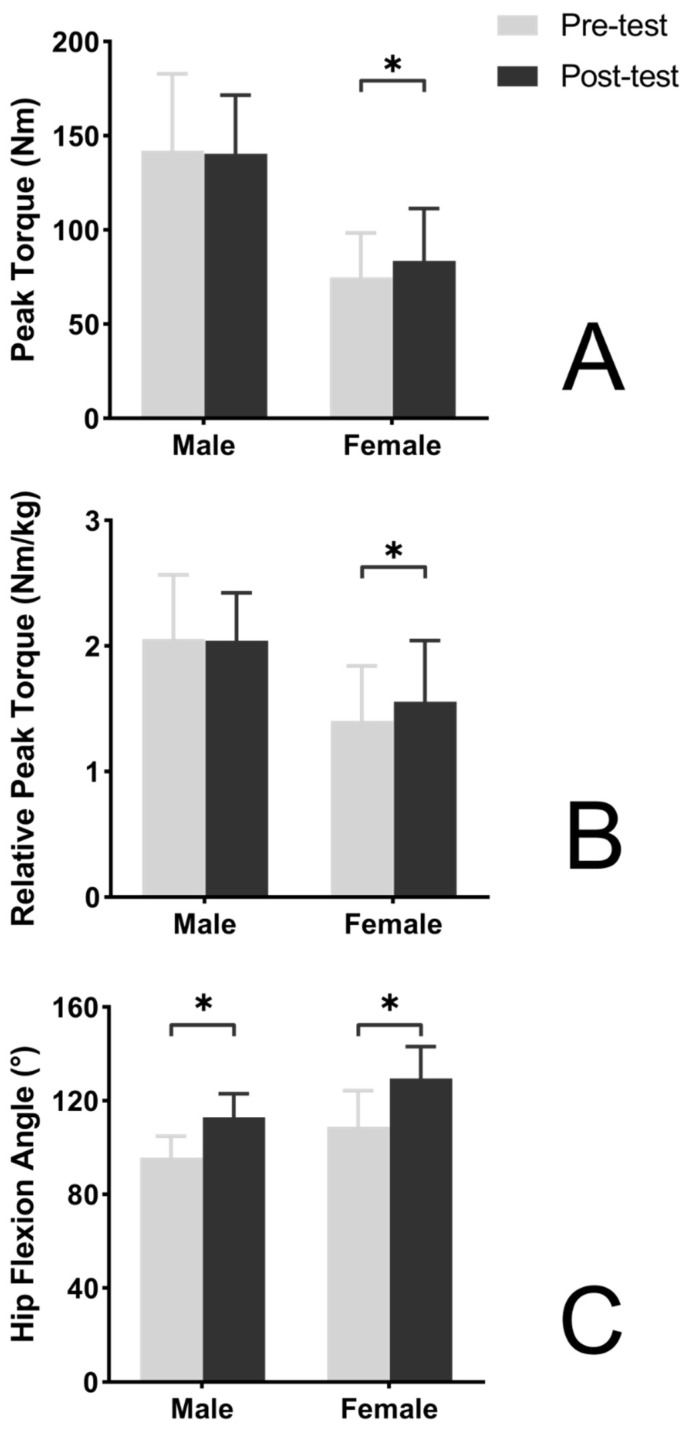
Effects of an 8-week flexibility training program on hamstring strength and flexibility. (**A**) Training effect on peak torque. (**B**) Training effect on relative peak torque. (**C**) Training effect on hip flexion angle. * Significant training effects.

**Table 1 ijerph-19-15256-t001:** Participant demographic characteristics (mean ± standard deviation).

Gender	Group	Number of Participants (N)	Age (Years)	Weight (kg)	Height (m)
Male	Flexibility	10	20.6 ± 1.6	70.5 ± 5.2	1.79 ± 0.04
	Strength	10	20.8 ± 2.0	64.0 ± 3.2	1.74 ± 0.03
Female	Flexibility	10	21.6 ± 1.4	55.6 ± 5.1	1.65 ± 0.05
	Strength	10	20.8 ± 1.5	57.1 ± 6.4	1.63 ± 0.05

Flexibility, flexibility training group. Strength, strength training group.

**Table 2 ijerph-19-15256-t002:** Detailed flexibility and strength training content.

Week	Flexibility Intervention			Strength Intervention		
	Content	Loading and Repetitions	Sets	Content	Loading and Repetitions	Sets
1	Walking knee lift	15 times	2	NHC with bend	8 reps	3
	Sitting toe touch	40 s/leg	2	Prone hamstrings curl	12 reps	4
	PNF stretch	50 s/leg	3	Physio-ball curl (two-leg)	8 reps	3
	Foam roll	40 s/leg	3	Glute bridge	50 s/leg	2
2–4	Forward lunge	15 times	2	NHC with bend	12 reps	3
	Sitting toe touch	50 s/leg	3	Prone hamstrings curl	14 reps	3
	PNF stretch	50 s/leg	3	Physio-ball curl (two-leg)	10 reps	3
	Foam roll	50 s/leg	3	Glute bridge	50 s/leg	2
5–8	Forward lunge	15 times	2	NHC	12 reps	3
	Semi-straddle	60 s/leg	2	Prone hamstrings curl	15 reps	3
	PNF stretch	50 s/leg	3	Physio-ball curl (two-leg)	10 reps/leg	2
	Foam roll	50 s/leg	4	Glute bridge	60 s/leg	2

PNF = Proprioceptive neuromuscular facilitation; NHC = Nordic hamstring curl. 30 s rest between sets, and 1 min rest between exercises.

## Data Availability

Not applicable.

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
