# Peer review of "Gender-Specific Effects of 8-Week Multi-Modal Strength and Flexibility Training on Hamstring Flexibility and Strength"

_ijerph, 2022, doi:10.3390/ijerph192215256_

Round 1

Reviewer 1 Report

The data has previously been presented in a paper "Effects of flexibility and strength interventions on optimal lengths of hamstring muscle-tendon units" Journal of Science and Medicine in Sport (2022). The authors have reduced the number of participants from the original data set raising questions around why some of the data has been omitted for this submission. This needs to be made clear and what this adds compared to your previous submission. 

Your previous paper has greater detail regards the methods used which have been omitted from the current submission. 

Author Response

We would like to thank all reviewers for the thorough review of our manuscript and constructive comments. We have carefully revised the manuscript according to reviewers’ suggestions. We hope the revised manuscript addressed all the concerns from the reviewers.

point 1: The data has previously been presented in a paper "Effects of flexibility and strength interventions on optimal lengths of hamstring muscle-tendon units" Journal of Science and Medicine in Sport (2022). The authors have reduced the number of participants from the original data set raising questions around why some of the data has been omitted for this submission. This needs to be made clear and what this adds compared to your previous submission.

Your previous paper has greater detail regards the methods used which have been omitted from the current submission.

Response 1: Data of this study was previously presented to determine the effects of altering hamstring flexibility or strength on hamstring optimal lengths. However, these two studies focused on two different aspects. In this manuscript, we sought to discuss whether multi-modal hamstring flexibility and strength training were gender influences. We didn’t reduce the number of participants as both studies used all the data from the 20 males and 20 females being recruited. We deleted some irrelevant test details like using Infrared reflective testing system to collect the three-dimensional (3-D) trajectories of reflective markers during isokinetic strength test, which aimed to measure hamstring optimal length in the previous study and are not pertinent to the purpose of present study. Furthermore, we added more testing details in Method. Hopefully this addresses the reviewer’s concern.

Author Response

We would like to thank all reviewers for the thorough review of our manuscript and constructive comments. We have carefully revised the manuscript according to reviewers’ suggestions. We hope the revised manuscript addressed all the concerns from the reviewers.

Point 1: Page 2, Lines 87-90: You need to expand on what you mean by regularly participated in exercise. Were these individuals who lifted weights, played in a softball league, or are just runners. This section may benefit from a list of inclusion and exclusion criteria. Also, explain how the subjects were randomly assigned. Was it every other person was assigned to the same group. Was a pre-randomized list created.

Response 1: Thank you for the comment. This paragraph was rewritten (Lines 103 - 110).

Point 2: Page 3, Line 113: What type of PNF stretching was performed (hold-relax, contract-relax, hold-relax-contract) and please explain it for people who may not be familiar with the concept.

Response 2: Thank you for the comment. Relevant details are provided in the revised manuscript (Lines 139 - 149).

Point 3: Page 3-4, Lines 119-129: You state that a camera was used to record body position, but do not explain how the angle was measured. Please include this explanation.

Response 3: Thank you for the comment. Relevant details are provided (Lines 190-193). We hope this addressed the reviewer’s concern.

Point 4: Page 4, Line 136-137: You state a isokinetic speed of 10°/s was used. Please explain the rationale for using this speed. Common isokinetic speeds for testing are 60°/s and 180°/s because these speeds have been found to reach the maximal capacities for force and power within a muscle.

Response 4: Although 60°/s and 180°/s are commonly used for isokinetic testing speeds, it has been suggested in literature that muscle tension is proportional to contraction time, with longer contraction time lead to greater tension. Since the isokinetic test in the current study was to evaluate maximal strength, we choose slower testing speed (10°/s) with the purpose to elicit maximal testing speed. Below are some references. Hopefully this address our concerns.

  1. Kabacinski J, Szozda PM, Mackala K, Murawa M, Rzepnicka A, Szewczyk P, Dworak LB. Relationship between Isokinetic Knee Strength and Speed, Agility, and Explosive Power in Elite Soccer Players. Int J Environ Res Public Health. 2022 Jan 7;19(2):671. doi: 10.3390/ijerph19020671. PMID: 35055489; PMCID: PMC8775831.
  2. Espinosa SE, Costello KE, Souza RB, Kumar D. Lower knee extensor and flexor strength is associated with varus thrust in people with knee osteoarthritis. J Biomech. 2020 Jun 23;107:109865. doi: 10.1016/j.jbiomech.2020.109865. Epub 2020 May 30. PMID: 32517867; PMCID: PMC7322620.

Point 5: Page 4, Line 159-163: Please include the use of Paired t-test for post-hoc analysis and how you adjusted for multiple comparisons in relation to significance.

Response 5: Thank you for the comment. No adjustment was used for multiple comparisons, as all post-hoc analyses were performed for variables that had two levels. Relevant details are provided (Line 208, Line 223). We hope this addressed the reviewer’s concern.

Reviewer 3 Report

Introduction

36 – may want to reword slightly (A study completed by Santos et al…)

Line 48 – please could you add in how much of a decrease was seen, would be good to know.

Line 51 – please could you add in how much of a decrease was seen, would be good to know.

Line 59 – have started to talk about aspects of rehabilitation. Whilst I agree in what you are saying, it doesn’t sit well necessarily with the theme of work you are discussing. Saying what you have already between lines 58 and start of line 59 is more than enough.

Line 69 – please could you add in how much of a decrease was seen, would be good to know.

Line 79-80 – typically HSI are seen in max velocity effort sprinting. The rationale to look at this is fair but might be a limitation in regards to the participants who you are using.

Methods

Please state exactly what does done in the warm up for both groups. Are they different?

Really good to see the sessions designed included and the strength session is designed logically. Only thing I would maybe include is the recovery times in between each sets. That will dictate some if the adaptations seen in individuals.

Could do with explaining the rationale as to how you progressed the sessions by each stage. What was the rationale to include an extra 10 seconds of stretching? Why has sets decreased in the final stages of the intervention?

Results

All fine

Discussion

Line 216 and 217 – you discussed %1RM which I agree with but because of the exercises you have chosen, it is difficult to determine exactly what %1RM your participants have performed at. Especially with the high repetition ranges you could call into question whether what you have done was truly strength training.

Line 270 – this statement around hypertrophy and stretching is questionable. I would direct you to work completed by BJ Schoenfeld and around the 3 mechanisms needed for hypertrophy

Lines 289-306 – you can also state that typically females may not engage in hamstring strength training such as Nordics. The increase in strength could be explained because it is just a new stimulus!

Some grammatical errors with authors name in reference list. May be a formatting issue.

Overall, this study is novel and will be helpful to a number of practitioners in the field. The suggestions made are minor and there is nothing major in my view that needs amending. Everyone associated with the study has done an excellent job on this and should be congratulated. 

Greg Henry - Hartpury University

Author Response

We would like to thank all reviewers for the thorough review of our manuscript and constructive comments. We have carefully revised the manuscript according to reviewers’ suggestions. We hope the revised manuscript addressed all the concerns from the reviewers.

Point 1: 36 – may want to reword slightly (A study completed by Santos et al…)

Response 1: Thank you for the comment. This paragraph was rewritten (Lines 39 - 40).

Point 2: Line 48 – please could you add in how much of a decrease was seen, would be good to know.

Response 2: Thank you for the comment. This paragraph was rewritten (Line 55).

Point 3: Line 51 – please could you add in how much of a decrease was seen, would be good to know.

Response 3: Thank you for the comment. This paragraph was rewritten (Lines 60-61).

Point 4: Line 59 – have started to talk about aspects of rehabilitation. Whilst I agree in what you are saying, it doesn’t sit well necessarily with the theme of work you are discussing. Saying what you have already between lines 58 and start of line 59 is more than enough.

Response 4: Thank you for the comment. This paragraph was deleted (Lines 68-69)..

Point 5: Line 69 – please could you add in how much of a decrease was seen, would be good to know.

Response 5: Thank you for the comment. This paragraph was rewritten (Line 79).

Point 6: Line 79-80 – typically HSI are seen in max velocity effort sprinting. The rationale to look at this is fair but might be a limitation in regards to the participants who you are using.

Response 6: Thank you for the comment. This paragraph was rewritten (Lines 91 - 93). We hope this addressed the reviewer’s concern.

Point 7: Please state exactly what does done in the warm up for both groups. Are they different?

Response 7: Thank you for the comment. The warm-up protocols were the same in both groups, and this paragraph was rewritten (Line 133).

Point 8: Really good to see the sessions designed included and the strength session is designed logically. Only thing I would maybe include is the recovery times in between each sets. That will dictate some if the adaptations seen in individuals.

Response 8: Thank you for the comment. This information was added in Table 2. (Lines 541 - 542).

Point 9: Could do with explaining the rationale as to how you progressed the sessions by each stage. What was the rationale to include an extra 10 seconds of stretching? Why has sets decreased in the final stages of the intervention?

Response 9: At the beginning of the training, the actual time for the subjects to complete the training may be relatively long because they are not familiar with the content and the program is therefore relatively difficult. In the middle and late stages of training, as a result of the gradual familiarization and adaptation to the training, the actual time for the subjects to complete the training will be relatively shortened, so the stretching time is appropriately adjusted to ensure that the entire training time is 30 minutes and this paragraph was rewritten (Line 129). Physio-ball curl action may be the action you mentioned that the sets decreased. The first four weeks are two-leg training at the same time, and the last four weeks are single-leg training. So the number of single-leg sets decreased and the actual training volume increased relatively. Thank you for the comment, the table has been added the leg information to clear the ambiguity (Table 2).

Point 10: Line 216 and 217 – you discussed %1RM which I agree with but because of the exercises you have chosen, it is difficult to determine exactly what %1RM your participants have performed at. Especially with the high repetition ranges you could call into question whether what you have done was truly strength training.

Response 10: The content of 1 RM was mainly used to demonstrate that the improvement of flexibility caused by strength training may be intensity-dependent, and 1 RM is not discussed in this study. This paragraph was rewritten (Line 276). We hope this addressed the reviewer’s concern.

Point 11: Line 270 – this statement around hypertrophy and stretching is questionable. I would direct you to work completed by BJ Schoenfeld and around the 3 mechanisms needed for hypertrophy

Response 11: Thank you for the comment. The content of muscle hypertrophy has been deleted (Lines 339-342). We hope this addressed the reviewer’s concern.

Point 12: Lines 289-306 – you can also state that typically females may not engage in hamstring strength training such as Nordics. The increase in strength could be explained because it is just a new stimulus!

Response 12: Thank you for the comment. The paragraph aimed to explain the reason for the strength increase for females after eight weeks flexibility training, and the flexibility protocol employed in the current study didn’t include Nordics training.

Round 2

Reviewer 2 Report

Thank you for making the stated corrections. I do not feel anything more is needed.